# Predictive Uncertainty Quantification for Graph Neural Network Driven Relaxed Energy Calculations

Joseph Musielewicz[1], Janice Lan[2], and Matt Uyttendaele[2]

[1]*Department of Chemical Engineering, Carnegie Mellon University*
[2]*Fundamental AI Research, Meta Platforms, Inc., Menlo Park, CA*

## Abstract

Graph neural networks (GNNs) have been shown to be astonishingly capable models for molecular property prediction, particularly as surrogates for expensive density functional theory calculations of relaxed energy for novel material discovery. However, one limitation of GNNs in this context is the lack of useful uncertainty prediction methods, as this is critical to the material discovery pipeline. In this work, we show that uncertainty quantification for relaxed energy calculations is more complex than uncertainty quantification for other kinds of molecular property prediction, due to the effect that structure optimizations have on the error distribution. We propose that distribution-free techniques are more useful tools for assessing calibration, recalibrating, and developing uncertainty prediction methods for GNNs performing relaxed energy calculations. We also develop a relaxed energy task for evaluating uncertainty methods for equivariant GNNs, based on distribution-free recalibration and using the Open Catalyst Project dataset. We benchmark a set of popular uncertainty prediction methods on this task, and show that latent distance methods, with our novel improvements, are the most well-calibrated and economical approach for relaxed energy calculations. Further, we challenge the community to develop improved uncertainty prediction methods for GNN-driven relaxed energy calculations, and benchmark them on this task.

## 1  Introduction

To keep up with growing energy demands, it is necessary to search for novel catalyst materials to enable more efficient storage of renewable sources of energy [1, 2, 3, 4]. Computational material discovery is crucial to this process, as it enables less expensive screening of an enormous space of possible catalyst materials than physical experiments. Faster and more accurate computational material discovery methods will be required to meet our society's renewable energy needs in the face of a rapidly changing climate.

Graph neural networks are state of the art in accelerating computational material discovery pipelines with machine learning potentials. Machine learning potentials work as surrogate models trained to approximate computationally expensive density functional theory (DFT) calculations of energy and forces on atomistic structures. This task is referred to as structure to energy and forces (S2EF). These energy and force calculations are used to iteratively perform geometric optimizations of atomic positions (referred to in this work as "relaxations"), to minimize their energy. These relaxed structure and relaxed energy calculations are what enable high-throughput predictions of catalyst performance in the real world. For a given catalyst-adsorbate system, the global minimum relaxed energy (adsorption energy) directly correlates with the reactivity and selectivity of reaction pathways on that catalyst surface [5, 6, 7, 8, 9, 10].

NeurIPS 2021 AI for Science Workshop.

In recent years, graph neural network (GNN)s have made tremendous strides in replacing DFT codes with an inexpensive, accurate alternative [11, 12, 13, 14]. Thanks to methods like AdsorbML, GNNs can speed up adsorption energy calculations alone at the cost of accuracy, or in tandem with DFT at the cost of speed [15]. The recent OCP Demo (https://open-catalyst.metademolab.com) is a publicly available tool where GNNs are used to calculate these adsorption energies without any expensive DFT calculations. However, a major limitation of current GNNs is their lack of uncertainty estimates for relaxed energy predictions. Ideally, users of these methods would know when it is safe to trust the GNN predictions, and when additional DFT calculations are warranted. In this work, we specifically examine methods of uncertainty quantification (UQ) of GNN predictions for this relaxed structure to relaxed energy (RS2RE) task.

## 2 Background

### 2.1 AdsorbML

AdsorbML [15] is a method to calculate adsorption energy using machine learning potentials. In order to find the global minimum relaxed energy for a specified surface and adsorbate, this method places the adsorbate in many different starting configurations, relaxes each configuration, and returns the minimum of all the relaxed energies. Traditionally, this would be done using an *ab initio* method such as DFT, but DFT is very costly and this approach is infeasibly expensive. AdsorbML uses GNNs as a surrogate for DFT to perform the relaxations instead, requiring only a solitary DFT single point calculation for the relaxed structure to verify the relaxed energy. The required number of expensive DFT calculations is further reduced by using GNNs to filter out all but the few most promising candidates. This method provides an adjustable spectrum of trade-offs between accuracy and efficiency, with one balanced option finding an equivalent or better adsorption energy 87.36% of the time while reducing DFT compute by more than a factor of 2000. Ideally, no DFT would be required, but even state of the art GNNs are unreliable energy predictors, and using them alone drops the success rate to 56%. Without DFT, such as in the OCP demo, we need uncertainty metrics, so users know when to trust the results of these models.

### 2.2 Graph Neural Networks

This work focuses on quantifying uncertainty prediction methods for EquiformerV2 [14], a GNN model architecture for molecular property prediction. We choose EquiformerV2 because it is the current state of the art in molecular property prediction for catalyst materials, according to the Open Catalyst Project (OCP) leaderboard[2]. We also compare it to Gemnet-OC [11], another high performing GNN on the leaderboard. Both of these models are used in the OCP Demo, to run the AdsorbML algorithm and predict minimum relaxed energies without the use of expensive DFT calculations[15].

### 2.3 Predictive Uncertainty Quantification

Many prior studies have examined the application of UQ techniques to machine learning potentials and molecular property prediction. Most UQ metrics seek to measure some description of the calibration of an uncertainty prediction method. The most popular UQ metrics are miscalibration area, Spearman's rank correlation coefficient, and the negative log likelihood of the errors given the uncertainties[16, 17, 18, 19, 20, 21, 22]. These metrics all rely on an assumption of of Gaussian errors, and all have significant drawbacks. Notably these metrics are not consistently in agreement about which uncertainty prediction performs best, even within a single study [23].

Calibration is the primary UQ metric for making direct comparisons between uncertainty prediction methods. Prior work by Rasmussen et al., Pernot, and Levi et al. show distribution free methods of measuring local and global miscalibration: the CI(Var(Z)) test and the error-based calibration plot [24, 25, 26]. These approaches can be more effective in describing the performance of predicted uncertainties of surrogate models when the expected error distribution cannot be assumed to be Gaussian. Error-based calibration measures can also be used to recalibrate uncertainty predictions, allowing a variety of uncertainty quantification methods to be recalibrated and compared.

# 3 Methods

## 3.1 Uncertainty Prediction Methods for GNNs

In this work we examine four common methods of uncertainty prediction on a pre-trained GNN: ensembles, latent space distances, mean variance estimation, and sequence regression models. The most prolific of these are ensemble methods, where a set of similar surrogate models are trained on similar sets of data to perform the same task, and the variance between their predictions is used to calculate the uncertainty of the model. We train ensembles of GNNs on the S2EF task to calculate energies of adsorbate-catalyst structures, but we are interested in the performance of predicting the uncertainty of EquiformerV2 on the relaxed structures. We construct three different ensembles for testing this approach, which we refer to as architecture (11 members), bootstrap (10 members), and parameter ensembles (6 members). Prior work has shown that diversity between members is typically the most important factor in developing a well-calibrated and expressive ensemble for uncertainty prediction [27, 28, 29, 30]. The architecture ensemble contains a variety of GNN model architectures, while the parameter and bootstrap ensembles contain only EquiformerV2 models, but vary the number of parameters, and the composition of training data respectively. More details on ensemble construction can be found in the supplemental information. Because we are specifically interested in the uncertainty of relaxed energy predictions, which requires a sequence of prior energy/force predictions during the relaxation, we hypothesize that taking the mean or the maximum of the predicted variances over each step (here referred to as a frame) of the trajectory might contain additional information which better models the uncertainty. For each ensemble composition, we test this theory by computing the uncertainty using the variance at the first frame, last frame, mean over all the frames, and max over over all the frames.

Another proven uncertainty method is the use of latent space distances [16, 17, 24]. In this approach, we extract some latent representation of each training point from the GNN, and create an index of these points to compute the L2-norm of the distance from any new test point to the training points. In practice, using libraries such as FAISS, the computational cost of this method is the lowest of any of the uncertainty methods we test [31]. Because this approach produces a distance of arbitrary scale, it is necessary to recalibrate on some calibration set to produce a meaningful uncertainty estimate. Prior work has shown the latent distance method to be effective for rotationally invariant models such as GemNet-OC, however EquiformerV2 improves upon the accuracy of GemNet-OC by preserving rotational equivariance all the way through the model [14, 11]. We expect this rotational equivariance will contribute undesirable noise to the L2 distance between latent representations, so we test this hypothesis by comparing the performance of the full EquiformerV2 latent representation, and the latent representation of its single rotationally invariant channel. We also compare these methods to the performance of using the latent space representation of GemNet-OC, trained on the same data, but used to predict the uncertainty of the same EquiformerV2 model. Additionally, for each latent distance approach, we test a novel strategy of computing the latent distance on a per-atom basis, and then taking the mean/max/sum over these distances. We compare this method to the more common approach of computing the latent distance for an entire frame by taking the mean of latent representations over all the atoms. More details on how all of the latent representations were extracted, and how the distances were computed can be found in the supplemental information.

The final categories of methods we test are mean variance estimation (MVE) and sequence regression models [22, 20, 23, 32]. In practice, we implement these methods in similar ways. For MVE methods we append an output-head, or an ensemble of output-heads to the EquiformerV2 architecture, and fine-tune this new output head on the calibration set to predict the residual of the energy prediction of the larger model. For the sequence regression model, we extract the same latent space representations used in the latent distance method, and train a sequence regression transformer architecture to predict the residual of the energy prediction on the calibration set. A significant distinction between these approaches is that the sequence regressor takes the latent representations of each frame of the trajectory as input, in sequence, with the hypothesis that some additional information about the uncertainty of the GNN model might be contained within its latent representations along the trajectory. More details on the implementation of both of these approaches can be found in the supplemental information.

## 3.2 Uncertainty Quantification Metrics

We hypothesize that the negative log likelihood (NLL), Spearman's rank correlation coefficient, and miscalibration area, which all assume a normal distribution of errors, will be inappropriate for uncertainty quantification on this task due to bias inherent to the RS2RE task. We test this using an approach suggested by Rasmussen et al. where we compute a simulated NLL and Spearman's coefficient by sampling from a normal distribution, with variance equal to the uncertainty, for each predicted uncertainty taken from an ensemble [24]. We perform this simulation 1000 times, and compute the average simulated metrics, then we compare this to the empirical NLL and Spearman's coefficient for the predicted uncertainty and measured error. If these metrics differ significantly from their simulated counterparts, then we infer that the errors do not follow the assumed normal distribution, and we consider these metrics to be ineffective for quantifying or calibrating these uncertainty methods.

Prior work by Pernot suggests the use of a distribution-free method to test whether an uncertainty method is calibrated [25]. The CI(Var(Z)) test uses the BCa boostrap method to compute a confidence interval of Var(Z) on a set of errors and uncertainties, without making any assumptions about the distribution [33, 34]. If 1 lies within the confidence interval, the uncertainty method is considered calibrated, because the Z values for a calibrated uncertainty metric are expected to have a variance of 1. Rasmussen et al. expands on this to suggest using an error-based calibration plot to quantify local calibration, distribution-free [26, 24]. The error-based calibration plot is predicated on the expected relationship between the root mean variance (RMV) and root mean square error (RMSE) being one-to-one.

$$RMSE = \sqrt{1/N_{bin} \sum_i \epsilon_i^2} \quad RMV = \sqrt{1/N_{bin} \sum_i \sigma_i^2} \quad \frac{RMSE}{RMV} \approx 1 \qquad (1)$$

We sort the test points by their predicted uncertainty, and then bin them into 20 bins. We compute the RMV of each bin, and the RMSE of each bin, using the BCa bootstrap method to compute a 95% confidence interval for the RMSE. Then we fit a line through the points to test for calibration, ideally the fitted line should have a high $R^2$ correlation with the points, and be as close as possible to the parity line. We can identify problems with local miscalibration where the parity line does not lie within the binned RMSE confidence intervals.

Table 1: Simulated NLL and Spearman correlations for uncertainty predictions made by identical ensembles across S2EF and RS2RE tasks. Values in parenthesis represent the standard deviation across 1000 runs of simulation.

| Method | Task | $NLL$ | $NLL_{sim}$ | $\rho$ | $\rho_{sim}$ | $A_{mis}$ | CI(Var(Z)) |
|---|---|---|---|---|---|---|---|
| bootstrap | RS2RE | 0.366 | -0.163(0.005) | 0.569 | 0.615(0.004) | 0.144 | [1.34, 2.14] |
| bootstrap | S2EF | 0.055 | -0.195(0.001) | 0.673 | 0.682(0.001) | 0.051 | [0.87, 0.90] |
| architecture | RS2RE | 0.380 | -0.117(0.005) | 0.604 | 0.610(0.004) | 0.136 | [1.25, 2.20] |
| architecture | S2EF | 0.075 | -0.097(0.001) | 0.670 | 0.641(0.001) | 0.023 | [0.68, 0.71] |

## 3.3 Recalibration and Evaluation

We compare all of the uncertainty methods benchmarked in this work after recalibration on a calibration set. In this case we use Open Catalyst 2020 dataset (OC20) in-domain validation set, and relax each structure (approximately 25,000 structures) to the same relaxation criterion as OC20, with the publicly available EquiformerV2 31M parameter checkpoint. We then compute a the Vienna Ab initio Simulation Package (VASP) single point calculation on the final frame to serve as the ground truth energy value for the RS2RE task [35, 36, 37, 38]. This set of relaxed energy predictions serves as the calibration set, and we repeat this process with the out-of-domain-both validation set to serve as the test set for this task. To recalibrate each uncertainty method, we use same the approach as the error-based calibration plot, to find the line of best fit through the binned RMSE/RMV points of the calibration set. We then simply recalibrate the uncertainty using the formula for the line of best fit:

$$\sigma_{recalibrated} = \text{slope}_{fit} * \sigma_{uncalibrated} + \text{intercept}_{fit} \qquad (2)$$

We suggest a small modification to the characterization of global calibration of uncertainties recalibrated with the error-based method. For this task, in addition to checking the global calibration

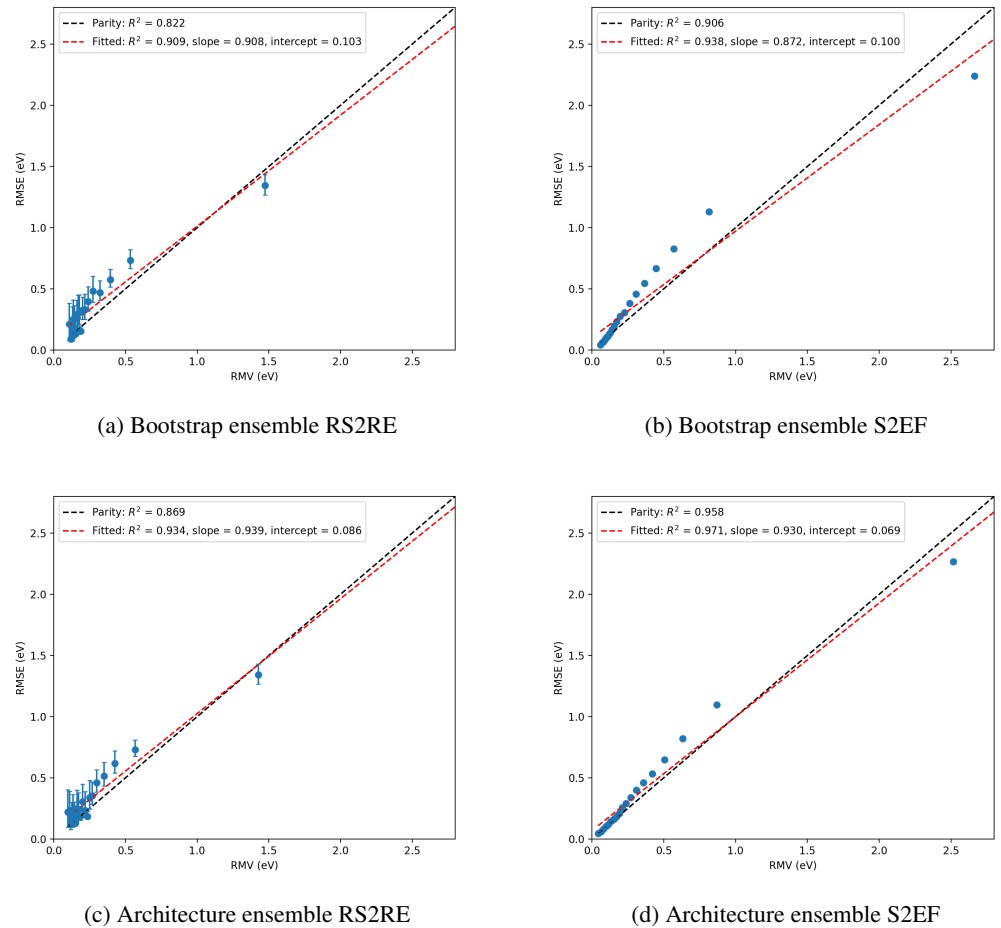

Figure 1: Error-based calibration plots on the out-of-domain test set for the two types of ensembles (bootstrap and architecture) and the two tasks (S2EF and RS2RE). With this distribution-free calibration measurement, we see that the bootstrap ensemble leads to a similar fit of the parity line in Figures 1a and 1b (i.e. the slope and intercept match more closely), while the architecture ensemble leads to a similar fit in Figures 1c and 1d. Note that the S2EF task contains more data, which causes the extremely narrow confidence intervals around the RMSE at each point, and the higher recorded errors.

CI(Var(Z)) test, we also select the most effective uncertainty metric by measuring the $R^2$ correlation of the binned RMSE/RMV ratio with the parity line. By recalibrating the uncertainty method on the calibration set and then testing on the test set, we can make direct comparisons between different uncertainty methods on the basis of their $R^2$ correlation with the parity line.

## 4 Results

### 4.1 Error Distribution and Uncertainty Quantification Metrics

Predicting the uncertainty of a surrogate model on the broader S2EF task is a much less challenging task than uncertainty prediction for RS2RE. We believe this is the result of selecting points only from the end of the structural optimization process creating a non-Gaussian distribution of errors from the predictions. This can be seen in Table 1, where the NLL and Spearman's rank correlation coefficientmetrics are very different from the simulated negative log likelihood ($NLL_{sim}$) and simulated Spearman's rank correlation coefficient ($Spearman_{sim}$) metrics for uncertainty predictions made by ensembles on the RS2RE task. Meanwhile, for the uncertainty predictions made on the S2EF

task, the NLL and Spearman's rank correlation coefficientmetrics are closer to their corresponding simulated metrics, although still too dissimilar to be considered Gaussian. This indicates that this assumption is closer to reality for the S2EF task than for the RS2RE task. Therefore metrics such as NLL, Spearman's rank correlation coefficient, and miscalibration area are much less appropriate for quantifying the performance of uncertainty prediction methods for the RS2RE task. Therefore we turn to distribution-free uncertainty quantification to understand which uncertainty prediction methods are most effective.

Using distribution-free uncertainty quantification in Figure 1, we see that the metrics are more similar across the RS2RE and RS2RE tasks for identical uncertainty prediction methods. The bootstrap ensemble is an overall worse fit on both tasks, and we can see it becomes less well calibrated earlier for both tasks. The architecture ensemble stays better calibrated for longer in both tasks. This is a good indicator of the distribution-free UQ techniques being more appropriate for characterizing uncertainty methods on this task, we expect to see similar behavior for the same method on similar tasks. This is unlike the UQ techniques which assume normally distributed errors, where the miscalibration area, NLL, and Spearman correlation coefficient report different comparisons of the same method across similar tasks. The larger quantity of data in the RS2RE task causes much smaller 95% confidence intervals to be computed by the bootstrap method for each bin. There is still some disparity between the two tasks, the RS2RE predictions are better calibrated globally according to the error-based calibration parity $R^2$. This is most likely a result of propagation of errors through the relaxation process being inherently difficult to account for by any method of uncertainty prediction.

## 4.2 Benchmarking Uncertainty Prediction Methods

Table 2: Distribution free calibration metrics for the best performing candidate uncertainty method for each of the categories we benchmark. Note that only the distance method is calibrated globally according to the CI(Var(Z)) test.

| Method | Parity $R^2$ | Fit $R^2$ | Slope | Intercept | CI(Var(Z)) |
|---|---|---|---|---|---|
| Latent distance | **0.952** | 0.967 | 1.022 | 0.026 | **[0.85, 1.58]*** |
| GNN ensemble | 0.905 | 0.940 | 0.946 | 0.070 | [1.09, 2.13] |
| Residual model | 0.331 | 0.955 | 1.200 | 0.127 | [2.64, 3.72] |
| MVE | 0.849 | 0.964 | 1.187 | 0.041 | [1.20, 1.97] |

Of the four methods we benchmark on the RS2RE task, the latent distance method is the best performer according to the distribution-free uncertainty quantification techniques, as seen in Table 2. The latent distance approach is the only method to pass the CI(Var(Z)) global calibration test, and it achieves the best calibration according to the parity $R^2$ on the out-of-domain test set. Note that all four models are capable of a comparable fit (if this test set were used for calibration) according to the similar scores on the fit $R^2$.

In figure 2 the error-based calibration plots and parity plots characterize the best performing latent distance method and ensemble. We see that the latent distance method generally retains good local calibration throughout, which we can see by the error bars of nearly every confidence interval intersect the parity line. Both methods do see poor calibration in the 0.6 eV to 1.0 eV range, but the ensemble method also experiences poorer calibration earlier than its competitor, starting near 0.3 eV. We see in the parity plots the latent distance method fans out in a more linear fashion, the parity line appears to be a good upper bound for the errors for longer than the ensemble method, which is desirable in a well-calibrated uncertainty method.

## 4.3 Comparing Distance Methods

The latent space representation sampled from EquiformerV2 after all graph convolutional interactions have been performed is inherently equivariant with respect to rotations. This is a valuable property for training GNN regressors to predict properties of molecular systems, but it renders distance measures invalid in the latent space, as latent space distance should be invariant and not equivariant to rotations of input structures. Since the same structure can be rotated to a numerically different latent space representation, many more samples must be present in training set for the latent space distance to

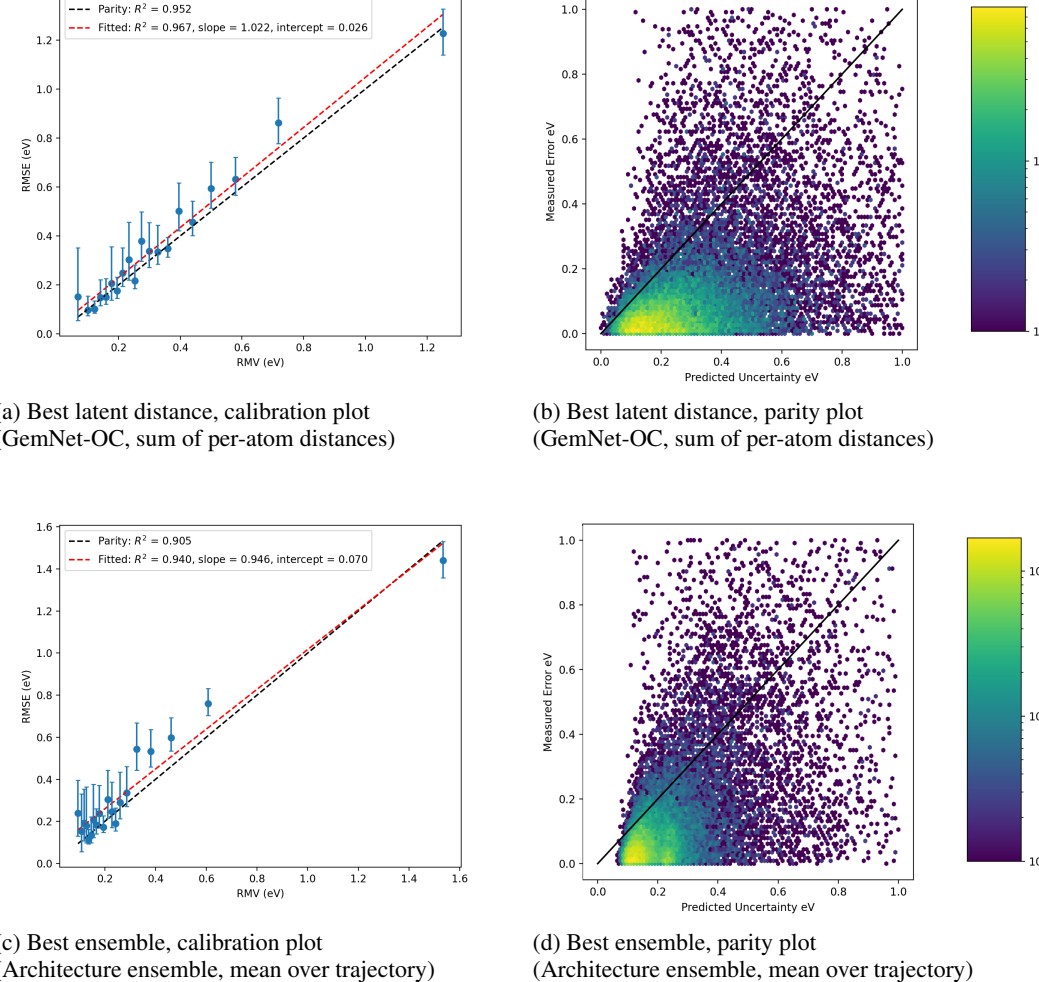

(a) Best latent distance, calibration plot
(GemNet-OC, sum of per-atom distances)

(b) Best latent distance, parity plot
(GemNet-OC, sum of per-atom distances)

(c) Best ensemble, calibration plot
(Architecture ensemble, mean over trajectory)

(d) Best ensemble, parity plot
(Architecture ensemble, mean over trajectory)

Figure 2: Parity plot and calibration for the best performing latent distance method and best performing ensemble method. The ensemble method shows worse global calibration, and suffers from poor local calibration at higher predicted uncertainties. The latent distance method shows better global and local calibration, with only slight local miscalibration in the second highest uncertainty bin.

be meaningful. Therefore we compare distances measured using the full latent space, to distances measured using only the rotationally invariant (degree 0 spherical harmonic) latent space.

Table 3 shows that the choice of latent space representation is by far the most important factor in choosing a good distance metric for predicting uncertainty. The second most important factor is choosing a per-atom distance metric, instead of taking the mean over all the atoms. And finally selecting the right type of per atom distance measure makes some difference, but is less significant as long as a good latent representation is chosen.

## 5 Conclusion

Effective uncertainty prediction methods for GNN relaxed energies are key to the development of faster and more accurate screening techniques for novel material discovery. Quantifying the performance of uncertainty methods on relaxed energy predictions is especially complex, due to distribution assumptions built into most commonly employed UQ techniques. Distribution-free techniques which employ bootstrapped confidence intervals, such as the CI(Var(Z)) test and error-based calibration plots, have been shown to be better metrics for analyzing the calibration

Table 3: Distribution free calibration metrics for each latent distance method tested. The latent representations correspond to model and method that was used to extract the latent representation for each atom. The distance corresponds to how the distance was computed for each system. A description of these methods can be found in section 3.1. A * indicates the method is calibrated according the CI(Var(Z)) test.

| Latent Rep. | Distance | Parity $R^2$ | Fit $R^2$ | Slope | Intercept | CI(Var(Z)) |
|---|---|---|---|---|---|---|
| EqV2 equiv. | atom max | -0.078 | 0.577 | 0.761 | 0.182 | [1.31, 1.67] |
| EqV2 equiv. | atom mean | **0.813** | 0.902 | 0.974 | 0.060 | [1.27, 180.65] |
| EqV2 equiv. | system mean | -14.709 | 0.000 | -0.236 | 0.547 | [1.29, 1.56] |
| EqV2 equiv. | atom sum | 0.672 | 0.907 | 0.711 | 0.159 | [1.24, 2.48] |
| EqV2 inv. | atom max | 0.842 | 0.892 | 0.983 | 0.054 | [0.98, 1.45]* |
| EqV2 inv. | atom mean | **0.866** | 0.913 | 0.974 | -0.037 | [0.90, 127178.90]* |
| EqV2 inv. | system mean | -0.628 | 0.795 | 1.540 | -0.061 | [1.38, 1.69] |
| EqV2 inv. | atom sum | 0.826 | 0.955 | 0.750 | 0.081 | [0.74, 1.35]* |
| GNOC | atom max | 0.924 | 0.967 | 0.951 | 0.064 | [0.94, 1.43]* |
| GNOC | atom mean | 0.932 | 0.965 | 1.108 | 0.006 | [0.87, 1.87]* |
| GNOC | system mean | 0.818 | 0.973 | 1.126 | 0.050 | [1.20, 1.66] |
| GNOC | atom sum | **0.952** | 0.967 | 1.022 | 0.026 | [0.85, 1.58]* |

of a UQ method in similar contexts, and we employ them here to great effect. We show that latent distance methods outperform ensembles and other uncertainty methods on the RS2RE task, which is of practical relevance to workflows such as AdsorbML. We also show that the choice of latent representation is very important to the calibration of the latent distance as an uncertainty metric. In the GNN latent space, atom-wise distances produce better calibrated than system-wise distances. Using rotationally invariant latent representations is crucial to producing calibrated distance measures, and the rotationally invariant latent space of GemNet-OC, a less accurate model, serves to compute a more well calibrated measure of uncertainty for EquiformerV2 than its own rotationally invariant latent space. Finally, we challenge the community to improve on this RS2RE task for predicting uncertainties, using our proposed recalibration framework as a measure. Future work in this area should also explore the prediction of global minimum energy uncertainties directly, and the development of model architectures training methods or distance measures which preserve rotational equivariance while producing meaningful latent space distances.

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

# Supporting Information

Joseph Musielewicz[1], Janice Lan[2], and Matt Uyttendaele[2]

[1]*Department of Chemical Engineering, Carnegie Mellon University*
[2]*Fundamental AI Research, Meta Platforms, Inc., Menlo Park, CA*

## A Ensemble Methods

The architecture ensemble contains eleven members, and uses several different graph neural network (GNN) architectures to achieve diversity, trained on largely the same data from the Open Catalyst 2020 dataset (OC20) structure to energy and forces (S2EF) training set. These model architectures include GemNet-OC, eSCN, and EquiformerV2. The parameters ensemble contains six members, all using the EquiformerV2 architecture, trained on the entire OC20 training set, and varies only by the number of parameters used during training. The bootstrap ensemble contains 10 identical members, using the EquiformerV2 architecture, each trained on a randomly selected slice (66%) of the OC20 training set. For each trajectory, we intend to predict a single uncertainty on the final frame, from the variance in the energy predictions of each of the members of the ensemble. We assess the predicted uncertainty for the ensemble methods by computing the variance in the energy predictions across all members of the ensemble for each system. We hypothesize that trajectory of all the energy predictions may contain some information relevant to predicting the uncertainty, so in addition to comparing across three ensembles, we also compare the effect of computing the uncertainty from the trajectory of variances. We consider four methods, using the variance of only the first frame, only the last frame, the max variance over the whole trajectory, and the mean of the variances over the whole trajectory. Taking the variances of all the frames over the whole trajectory contributes significantly more computational cost to this method.

Table 1: Distribution free calibration metrics for each ensemble method tested. We construct three different ensembles, and test them using three different methods of predicting the relaxed energy uncertainty. Although some are close, none of the ensembles passed the CI(Var(Z)) test.

| Ensemble | Method | Parity $R^2$ | Fit $R^2$ | Slope | Intercept | CI(Var(Z)) |
|---|---|---|---|---|---|---|
| Bootstrap | First | 0.809 | 0.896 | 1.056 | 0.065 | [1.14, 1.65] |
| Bootstrap | Last | 0.838 | 0.914 | 0.897 | 0.104 | [1.26, 2.00] |
| Bootstrap | Max | 0.817 | 0.890 | 1.025 | 0.074 | [1.19, 1.92] |
| Bootstrap | Mean | **0.870** | 0.930 | 0.912 | 0.094 | [1.20, 2.06] |
| Parameters | First | 0.793 | 0.874 | 0.985 | 0.081 | [1.15, 1.64] |
| Parameters | Last | 0.842 | 0.927 | 0.828 | 0.114 | [1.25, 1.99] |
| Parameters | Max | 0.844 | 0.905 | 0.978 | 0.079 | [1.11, 1.75] |
| Parameters | Mean | **0.849** | 0.921 | 0.845 | 0.107 | [1.16, 1.94] |
| Architecture | First | 0.830 | 0.933 | 1.050 | 0.070 | [1.20, 1.81] |
| Architecture | Last | 0.891 | 0.942 | 0.945 | 0.081 | [1.18, 2.03] |
| Architecture | Max | 0.860 | 0.933 | 1.078 | 0.053 | [1.09, 1.88] |
| Architecture | Mean | **0.905** | 0.940 | 0.946 | 0.070 | [1.09, 2.13] |

Table 1 shows the choice of ensemble seems to be the most important to the success of the uncertainty prediction, followed by the method used to predict the uncertainty. For the better ensemble (architecture) all but the first-frame method beat every other ensemble-method combination in terms of parity $R^2$. This supports prior work which shows diversity between different members of an ensemble to be

the most critical factor in producing good uncertainty predictions. Within each ensemble, there is a fairly consistent pattern of the mean method (taking the mean over the variances of all the frames in the trajectory) being the most effective method of computing uncertainty. This is followed by either the max method, or the last frame method, and then the first frame method is always the worst. This aligns with the notion that some additional information about the uncertainty of the model on a particular relaxed point can be gleaned from its uncertainty on other related points along the trajectory.

## B  Distance Methods

The distance methods make use of the distance between points in some hidden latent space for each inference call. In every case, we extract the latent representation from some hidden layer after the interaction blocks in a GNN. We then compute a latent space representation for every relaxed system in the training data set (460,000 systems), and relaxed systems from the in-domain and out-of-domain validation data sets (25,000 systems each). The in-domain validation data is used as a calibration set, while the out-of-domain validation data is used as the test set. For each data point in the calibration and test sets, we compute the shortest distance from that latent representation to any latent representation in the train set. These latent representations are calculated on a per-atom basis, with two options for computing distances. Either the distances between atom representations are computed, or the mean of the representations is computed and the distances between those means are computed. We make three different comparisons to assess the effects of different choices on the performance of latent space distance methods.

First we compare the effect of using the distance between the means of the latent representation over the entire system, to using the distances between the per-atom latent representations. For the per-atom latent representations, to reduce all the per-atom distances to a single value, we compare taking the mean of the distances, the sum of the distances, or the maximum of the distances. Across the different versions of the EquiformerV2 latent space that we sampled, we found that any version of the per-atom latent distances universally outperformed the per-system latent distance. Within the per-atom latent distance approaches, we found the mean of the per-atoms distances to be often the best performer, and reliably high-performing across all models and latent representations. These results can be found in the main text.

Second we compare the effect of different latent space sampling methods for expressing rotational equivariance and/or invariance in the latent representation. We do this by sampling latent representations from EquiformerV2 from certain spherical channels, and before and after edge alignment. We take three approaches to sampling spherical channels: sampling only the l=0, m=0 channel, sampling all the m=0 channels, and sampling all the channels. The l=0, m=0 channel should be inherently invariant to rotation. All of the m=0 channels should be sensitive to some rotations when the representation is subject to random rotations, but during the output blocks, the representation is reliably rotated to be aligned with the edges of the input graph, and therefore these channels should behave as though they are invariant to rotation. Finally, the representation of all of the channels should always be equivariant to rotation [1]. We find that using all the channels for the latent space is consistently worse than using only the invariant channels. And that using the edge-aligned version of the l=0, m=0 performs best. These results can be found in Table 2.

Table 2: Comparison of latent space distances extracted from Equiformer V2 using different approaches aimed at eliminating rotational equivariance, all used to predict the error of Equiformer V2 on relaxed structures.

| Model | Sphharms | Parity $R^2$ | Fit $R^2$ | Slope | Intercept | CI(Var(Z)) |
|---|---|---|---|---|---|---|
| EqV2 unaligned | (m=2, l=4) | 0.813 | 0.902 | 0.974 | 0.060 | [1.27, 180.65] |
| EqV2 aligned (sphharm) | (m=2, l=4) | 0.788 | 0.872 | 1.055 | 0.042 | [1.13, 2.79] |
| EqV2 aligned (channel) | (m=2, l=4) | 0.745 | 0.957 | 1.211 | 0.008 | [1.17, 2.02] |
| EqV2 unaligned | (m=0, l=0) | 0.866 | 0.913 | 0.974 | -0.037 | [0.90, 127178.90]* |
| EqV2 aligned | (m=0, l=0) | **0.910** | 0.911 | 1.014 | -0.002 | [1.43, 369.43] |
| EqV2 aligned | (m=0, l=4) | 0.906 | 0.913 | 0.998 | -0.020 | [0.92, 24.51]* |

Third, we test 4 different GNNs for their ability to predict the uncertainty of relaxed structure to relaxed energy (RS2RE) predictions made using EquiformerV2: PAINN, eSCN, EquiformerV2, and GNOC [2, 3, 4, 1]. PAINN, eSCN, and EquiformerV2 all take a rotationally equivariant approach, while GNOC preserves rotational invariance throughout the model. In each case, we sample the entire latent space of the GNN immediately after the final interaction block. We find that GNOC outperforms even the invariant latent space of EquiformerV2 at predicting EquiformerV2's uncertainty. These results can be found in Table 3.

Table 3: Comparison of latent space distances extracted from different models, all used to predict the error of Equiformer V2 on relaxed structures.

| Model | Distance | Parity $R^2$ | Fit $R^2$ | Slope | Intercept | CI(Var(Z)) |
|-------|----------|--------------|-----------|-------|-----------|------------|
| EqV2 equiv. | atom mean | 0.813 | 0.902 | 0.974 | 0.060 | [1.27, 180.65] |
| EqV2 inv. | atom mean | 0.866 | 0.913 | 0.974 | -0.037 | [0.90, 127178.90]* |
| eSCN | atom mean | 0.818 | 0.931 | 1.090 | 0.021 | [1.13, 5.82] |
| PAINN | atom mean | 0.856 | 0.934 | 0.971 | -0.050 | [0.57, 1.04]* |
| GNOC | atom mean | **0.932** | 0.965 | 1.108 | 0.006 | [0.87, 1.87]* |

## C   MVE and Sequence Regression Methods

The MVE and sequence regression methods we tested both aim to directly predict the uncertainty of EquiformerV2 by training a neural network, or portion of a neural network. These models take the full latent representation at the end of the last interaction block, and are fit on residuals of the EquiformerV2's energy predictions. In the case of the MVE methods, an additional output head, or an ensemble of output heads is added to the fully trained EquiformerV2 checkpoint. These new output heads are initialized randomly, the rest of the model is frozen, and they are trained on the residuals of the in-domain validation data, until the performance of the direct residual prediction stops improving on a held out portion of the validation data set. In the case of the single head, the loss is computed as a the difference between its direct prediction and the residual values. In the case of the ensemble of heads, the loss is computed as the difference between the variance in each of the ten heads energy predictions, and the residual values.

The sequence regression models are similarly trained using the full latent representations as input. However, we hypothesize that change in the latent representation over the entire trajectory might contain information relevant to the task of directly learning the residuals on the last frame. We use a transformer sequence regression model, as implemented in Hugging Face [5]. We modify the transformer to accept vectors of latent representations for each atom as input, and batch over all of the atoms in the system. Then for each trajectory, we train it on the sequence of atoms, batching over all the atoms, and regress to fit the residual on the final frame of the EquiformerV2 trajectory. Similar to the MVE approach, we train on the in-domain validation data set, until performance stops improving on a held out portion of the data set. As in all other cases, we recalibrate all uncertainty predictions on the out-of-domain validation data set, using error-based recalibration. We compare the best performing result for each of these implementations in Table 4. However we note that both of these direct residual fitting methods had a tendency to overfit, performing much more poorly on the out-of-domain validation data than on the in-domain validation data, despite using a held-out portion of the validation data to perform early stopping.

## D   Additional Results

### D.1   UMAP of EquiformerV2 Latent Spaces

In Figure 1 we see the results of performing UMAP dimensionality reduction on a rotationally equivariant and invariant versions of the EquiformerV2 latent space [6]. Dense clustering in the invariant latent space shows that all of the elements are easily distinguished. While the noisy clustering in the equivariant latent space, particularly the overlap between metals like Cu and Pd, shows that the equivariant latent space makes these distinctions less clear. This seems to align with explanations for why the distance methods for uncertainty perform better in the invariant versions

of these latent spaces, since the distances between different atoms ought to be less noisy and more meaningful.

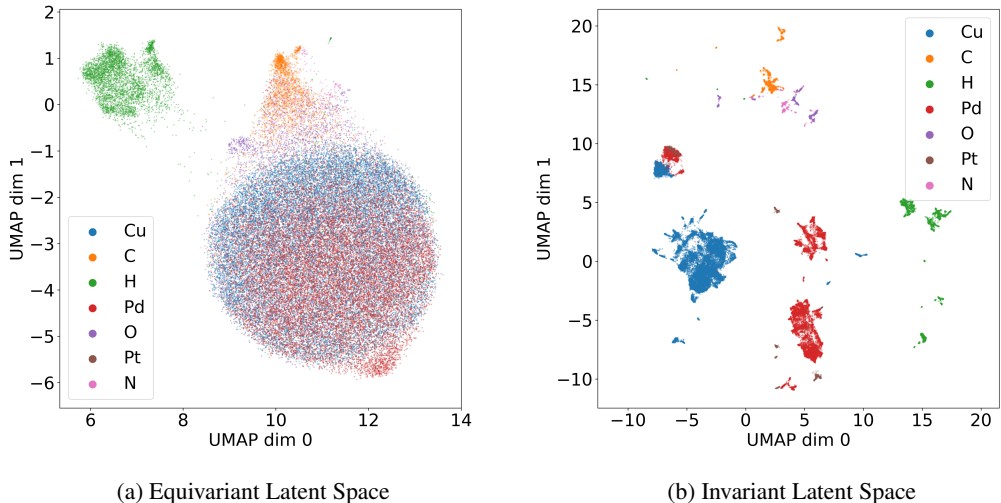

(a) Equivariant Latent Space

(b) Invariant Latent Space

Figure 1: Plots of UMAP dimensionality reduction performed on the equivariant (all channels) and invariant (m=0, l=0) latent spaces for a subset of the training set. We see that the different elements represented are clearly clustered in both plots, but that there is significantly more noise in the clustering of similar elements in the equivariant latent space, while the invariant latent space clusters are much denser and less noisy

## D.2 Error Distribution

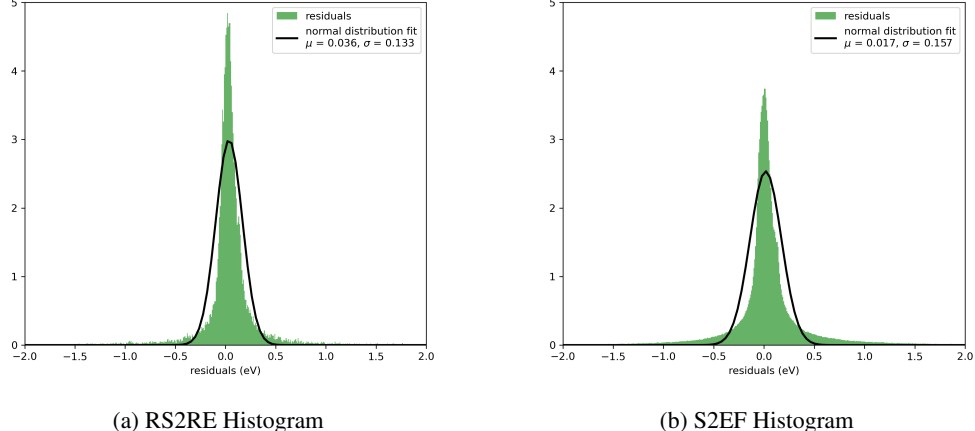

(a) RS2RE Histogram

(b) S2EF Histogram

Figure 2: Histograms of the distribution of errors for the bootstrap ensemble on S2EF and RS2RE tasks. The errors should follow the normal distribution, and we have plotted a normal distribution fit to each measured distribution. We can see that neither task truly follows a normal distribution, however it seems that the RS2RE task in (2a) is less aligned with the normal distribution than the S2EF task in (2b). This leads to worse miscalibration area despite the fact that the two tasks ought to be quite similar for the same uncertainty method.

## D.3 Comparing Best Methods per Metric

Table 4: Best performers for each of the UQ metrics. UQ metrics which assume normally distributed errors for a non-normal error distribution appear to be very inconsistent with one another. Only method that is calibrated according to the CI(Var(Z)) test is the distance method.

| Method | Parity $R^2$ | $A_{mis,u}$ | $A_{mis,r}$ | $NLL$ | $\rho$ | AUROC | CI(Var(Z)) |
|---|---|---|---|---|---|---|---|
| GNOC distance | **0.952** | 0.365 | 0.178 | 0.279 | 0.468 | 0.755 | **[0.85, 1.58]\*** |
| Architecture ens. | 0.830 | **0.050** | 0.156 | 0.411 | 0.453 | 0.746 | [1.20, 1.81] |
| Traj. transformer | 0.331 | 0.174 | **0.102** | 1.170 | 0.360 | 0.695 | [2.64, 3.72] |
| MVE head | 0.831 | 0.340 | 0.114 | **0.251** | 0.526 | 0.785 | [1.15, 1.89] |
| Bootstrap ens. | 0.870 | 0.248 | 0.121 | 0.303 | **0.585** | **0.817** | [1.20, 2.06] |

Table 5: Simulated NLL and Spearman coefficient scores for the best performers for each of the UQ metrics. Simulated metrics appear to suggest that the assumption of normally distributed errors is inaccurate for this RS2RE data.

| Method | Parity $R^2$ | $NLL$ | $NLL_{sim}$ | $\rho$ | $\rho_{sim}$ | CI(Var(Z)) |
|---|---|---|---|---|---|---|
| GNOC distance | 0.952 | 0.279 | 0.092(0.005) | 0.468 | 0.469(0.005) | [0.85, 1.58]\* |
| Architecture ens. | 0.830 | 0.411 | -0.003(0.004) | 0.453 | 0.640(0.004) | [1.20, 1.81] |
| Traj. transformer | 0.331 | 1.170 | -0.243(0.004) | 0.360 | 0.554(0.005) | [2.64, 3.72] |
| MVE head | 0.831 | 0.251 | -0.187(0.004) | 0.526 | 0.584(0.004) | [1.15, 1.89] |
| Bootstrap ens. | 0.870 | 0.303 | -0.163(0.004) | 0.585 | 0.603(0.004) | [1.20, 2.06] |

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
