# OpenReview forum: "Predictive Uncertainty Quantification for Graph Neural Network Driven Relaxed Energy Calculations"
_NeurIPS.cc/2023/Workshop/AI4Science — NeurIPS2023-AI4Science Poster_

### Official Review · Reviewer_T5Da · 2023-10-24

**Rating:** 7
**Confidence:** 3

**Review:**

**Summary**
This paper benchmarks uncertainty quantification methods for GNN relaxed energy prediction. Models, metrics and tasks are explored in the paper.


**Strengths and Weakness:**
The benchmark models are diverse and extensive: four uncertainty quantification methods: 1. Ensembles 2. latent space distance 3. mean variance estimation and 4. sequence regression models.
The metrics are carefully chosen and the calibration is discussed with experiments. Recalibration is then used for the benchmarking of models (Distribution-free calibration metrics)
The details of the ensembles and parameters are not listed so it is hard to judge which ones are being benchmarked (although it mentions “more information on the construction of any of the ensembles can be found in the SI” but there is no such information)

**Minor:**
	Table1 in Supplement, missing section number

**Conclusion:**
This work extensively benchmark and discuss the uncertainty quantification methods for GNN relaxed energy prediction. Different aspects such as model, metrics and tasks are considered. The paper is well-written and the results are supported by the experiments. Although the work itself is for a specific GNN task (predicting energy from structure),, the methods and discussion in the paper could be inspiring for other related graph tasks. Accept is recommended.

---

### Official Review · Reviewer_pAVA · 2023-10-25
**An interesting paper to study uncertainty quantification for GNNs in material discovery**

**Rating:** 6
**Confidence:** 3

**Review:**

### Summary of the paper
This paper benchmarks four common uncertainty quantification (UQ) methods for GNNs used for relaxed energy calculations. With extensive experiments, a few observations have been made to shed light on future research.


### Strengths
- The paper is well-motivated and studies a very interesting and critical direction of UQ for GNNs in material discovery.
- Extensive experiments have been conducted to benchmark various UQ methods for this task, which empirically justifies the superiority of distribution-free techniques and serves as great baselines for future studies.
- The experiments, in particular, have shown latent distance methods may hold great promise for this task. The authors further explored this type of method and proposed to use atom-wise distances for better performance.

### Weaknesses
- The study mainly focuses on EquiformerV2. It would be great to also have extensive studies across different backbones to see if the observations may hold for diverse cases.
- Though validated empirically, principled analysis can strengthen the observations of the paper.

### Summary of the review
Overall, I find this paper studies a critical gap in the literature by focusing on UQ for GNNs in the context of relaxed energy calculations. The benchmarking results and observations may facilitate the development of this area.

---

### Meta-Review · Area_Chair_Lou1 · 2023-10-27

**Recommendation:** Accept (Poster)
**Confidence:** 5

**Metareview:**

This paper studies the uncertainty in the geometric modeling for energy and force prediction, and it can provide us with a high-level idea of how much we can trust the ML models. This work carefully scrutinizes four common UQ methods with valid evaluation metrics, and it opens a new direction on studying UQ in this research line. Authors please also fix the minor issues proposed by the reviewers.